# Fully Understanding The Hashing Trick

**Casper Freksen** *
Department of Computer Science
Aarhus University, Denmark
`cfreksen@cs.au.dk`

**Lior Kamma** *
Department of Computer Science
Aarhus University, Denmark
`lior.kamma@cs.au.dk`

**Kasper Green Larsen** *
Department of Computer Science
Aarhus University, Denmark
`larsen@cs.au.dk`

## Abstract

Feature hashing, also known as *the hashing trick*, introduced by Weinberger *et al.* (2009), is one of the key techniques used in scaling-up machine learning algorithms. Loosely speaking, feature hashing uses a random sparse projection matrix $A : \mathbb{R}^n \to \mathbb{R}^m$ (where $m \ll n$) in order to reduce the dimension of the data from $n$ to $m$ while approximately preserving the Euclidean norm. Every column of $A$ contains exactly one non-zero entry, equals to either $-1$ or $1$.

Weinberger *et al.* showed tail bounds on $\|Ax\|_2^2$. Specifically they showed that for every $\varepsilon, \delta$, if $\|x\|_\infty / \|x\|_2$ is sufficiently small, and $m$ is sufficiently large, then

$$\Pr[\; |\; \|Ax\|_2^2 - \|x\|_2^2 \;| < \varepsilon \|x\|_2^2 \;] \geq 1 - \delta \;.$$

These bounds were later extended by Dasgupta *et al.* (2010) and most recently refined by Dahlgaard *et al.* (2017), however, the true nature of the performance of this key technique, and specifically the correct tradeoff between the pivotal parameters $\|x\|_\infty / \|x\|_2, m, \varepsilon, \delta$ remained an open question.

We settle this question by giving tight asymptotic bounds on the exact tradeoff between the central parameters, thus providing a complete understanding of the performance of feature hashing. We complement the asymptotic bound with empirical data, which shows that the constants "hiding" in the asymptotic notation are, in fact, very close to 1, thus further illustrating the tightness of the presented bounds in practice.

## 1 Introduction

*Dimensionality reduction* that approximately preserves Euclidean distances is a key tool used as a preprocessing step in many geometric, algebraic and classification algorithms, whose performance heavily depends on the dimension of the input. Loosely speaking, a distance-preserving dimensionality reduction is an (often random) embedding of a high-dimensional Euclidean space into a space of low dimension, such that pairwise distances are approximately preserved (with high probability). Its applications range upon nearest neighbor search [AC09, HIM12, AIL+15], classification and regression [RR08, MM09, PBMID14], manifold learning [HWB08] sparse recovery [CT06] and numerical linear algebra [CW09, MM13, Sár06]. For more applications see, e.g. [Vem05].

One of the most fundamental results in the field was presented in the seminal paper by Johnson and Lindenstrauss [JL84].

**Lemma 1** (Distributional JL Lemma). *For every $n \in \mathbb{N}$ and $\varepsilon, \delta \in (0, 1)$, there exists a random $m \times n$ projection matrix $A$, where $m = \Theta(\varepsilon^{-2} \lg \frac{1}{\delta})$ such that for every $x \in \mathbb{R}^n$*

$$\Pr[\, |\, \|Ax\|_2^2 - \|x\|_2^2 \,| < \varepsilon \|x\|_2^2 \,] \geq 1 - \delta \qquad (1)$$

The target dimension $m$ in the lemma is known to be optimal [JW13, LN17].

**Running Time Performances.** Perhaps the most common proof of the lemma (see, e.g. [DG03, Mat08]) samples a projection matrix by independently sampling each entry from a standard Gaussian (or Rademacher) distribution. Such matrices are by nature very dense, and thus a naïve embedding runs in $O(m\|x\|_0)$ time, where $\|x\|_0$ is the number of non-zero entries of $x$.

Due to the algorithmic significance of the lemma, much effort was invested in finding techniques to accelerate the embedding time. One fruitful approach for accomplishing this goal is to consider a distribution over *sparse* projection matrices. This line of work was initiated by Achlioptas [Ach03], who constructed a distribution over matrices, in which the *expected* fraction of non-zero entries is at most one third, while maintaining the target dimension. The best result to date in constructing a sparse Johnson-Lindenstrauss matrix is due to Kane and Nelson [KN14], who presented a distribution over matrices satisfying (1) in which every column has at most $s = O(\varepsilon^{-1} \lg(1/\delta))$ non-zero entries. Conversely Nelson and Nguyễn [NN13] showed that this is almost asymptotically optimal. That is, every distribution over $n \times m$ matrices satisfying (1) with $m = \Theta(\varepsilon^{-2} \lg(1/\delta))$, and such that every column has at most $s$ non-zero entries must satisfy $s = \Omega((\varepsilon \lg(1/\varepsilon))^{-1} \lg(1/\delta))$.

While the bound presented by Nelson and Nguyễn is theoretically tight, we can provably still do much better in practice. Specifically, the lower bound is attained on vectors $x \in \mathbb{R}^n$ for which, loosely speaking, the "mass" of $x$ is concentrated in few entries. Formally, the ratio $\|x\|_\infty / \|x\|_2$ is large. However, in practical scenarios, such as the term frequency - inverse document frequency representation of a document, we may often assume that the mass of $x$ is "well-distributed" over many entries (That is, $\|x\|_\infty / \|x\|_2$ is small). In these common scenarios projection matrices which are significantly sparser turn out to be very effective.

**Feature Hashing.** In the pursuit for sparse projection matrices, Weinberger *et al.* [WDL+09] introduced dimensionality reduction via *Feature Hashing*, in which the projection matrix $A$ is, in a sense, as sparse as possible. That is, every column of $A$ contains exactly one non-zero entry, randomly chosen from $\{-1, 1\}$. This techniqueis one of the most influencial mathematical tools in the study of scaling-up machine learning algorithms, mainly due to its simplicity and good performance in practice [Dal13, Sut15]. More formally, for $n, m \in \mathbb{N}^+$, the projection matrix $A$ is sampled as follows. Sample $h \in_R [n] \rightarrow [m]$, and $\sigma = \langle \sigma_j \rangle_{j \in [n]} \in_R \{-1, 1\}^n$ independently. For every $i \in [m], j \in [n]$, let $a_{ij} = a_{ij}(h, \sigma) := \sigma_j \cdot \mathbb{1}_{h(j)=i}$ (that is, $a_{ij} = \sigma_j$ iff $h(j) = i$ and 0 otherwise). Weinberger *et al.* additionally showed exponential tail bounds on $\|Ax\|_2^2$ when the ratio $\|x\|_\infty / \|x\|_2$ is sufficiently small, and $m$ is sufficiently large. These bounds were later improved by Dasgupta *et al.* [DKS10] and most recently by Dahlgaard, Knudsen and Thorup [DKT17a] improved these concentration bounds. Conversely, a result by Kane and Nelson [KN14] implies that if we allow $\|x\|_\infty / \|x\|_2$ to be too large, then there exist vectors for which (1) does not holds.

Finding the correct tradeoffs between $\|x\|_\infty / \|x\|_2$, and $m, \varepsilon, \delta$ in which feature hashing performs well remained an open problem. Our main contribution is settling this problem, and providing a complete and comprehensive understanding of the performance of feature hashing.

## 1.1 Main results

The main result of this paper is a tight tradeoff between the target dimension $m$, the approximation ratio $\varepsilon$, the error probability $\delta$ and $\|x\|_\infty / \|x\|_2$. More formally, let $\varepsilon, \delta > 0$ and $m \in \mathbb{N}^+$. Let $\nu(m, \varepsilon, \delta)$ be the maximum $\nu \in [0, 1]$ such that for every $x \in \mathbb{R}^n$, if $\|x\|_\infty \leq \nu \|x\|_2$ then (1) holds. Our main result is the following theorem, which gives tight asymptotic bounds for the performance of feature hashing, thus closing the long-standing gap.

**Theorem 2.** *There exist constants $C \geq D > 0$ such that for every $\varepsilon, \delta \in (0,1)$ and $m \in \mathbb{N}^+$ the following holds. If $\frac{C \lg \frac{1}{\delta}}{\varepsilon^2} \leq m < \frac{2}{\varepsilon^2 \delta}$ then*

$$\nu(m, \varepsilon, \delta) = \Theta \left( \sqrt{\varepsilon} \cdot \min \left\{ \frac{\lg \frac{\varepsilon m}{\lg \frac{1}{\delta}}}{\lg \frac{1}{\delta}}, \sqrt{\frac{\lg \frac{\varepsilon^2 m}{\lg \frac{1}{\delta}}}{\lg \frac{1}{\delta}}} \right\} \right) .$$

*Otherwise, if $m \geq \frac{2}{\varepsilon^2 \delta}$ then $\nu(m, \varepsilon, \delta) = 1$. Moreover if $m < \frac{D \lg \frac{1}{\delta}}{\varepsilon^2}$ then $\nu(m, \varepsilon, \delta) = 0$.*

While the bound presented in the theorem may strike as surprising, due to the intricacy of the expressions involved, the tightness of the result shows that this is, in fact, the correct and "true" bound. Moreover, the proof of the theorem demonstrates how both branches in the $\min$ expression are required in order to give a tight bound.

**Experimental Results.** Our theoretical bounds are accompanied by empirical results that shed light on the nature of the constants in Theorem 2. Our empirical results show that in practice the constants inside the $\Theta$-notation are significantly tighter than the theoretical proof might suggest, and in fact feature hashing performs well for a larger scope of vectors. Specifically, for a synthetic set of generated bit-vectors, we show that whenever $\frac{4 \lg \frac{1}{\delta}}{\varepsilon^2} \leq m < \frac{2}{\varepsilon^2 \delta}$ the constant hidden by the $\Theta$-notation is at least $0.75$ (except for very sparse vectors, i.e. $\|x\|_0 \leq 7$). That is

$$\nu(m, \varepsilon, \delta) \geq 0.725 \sqrt{\varepsilon} \cdot \min \left\{ \frac{\lg \frac{\varepsilon m}{\lg \frac{1}{\delta}}}{\lg \frac{1}{\delta}}, \sqrt{\frac{\lg \frac{\varepsilon^2 m}{\lg \frac{1}{\delta}}}{\lg \frac{1}{\delta}}} \right\} .$$

For a bag-of-words representation of 1500 NIPS papers with stopwords removed [DKT17b, New08] our experiments show that the constant is even larger, whereas the theoretical proof provides a much smaller constant of $2^{-6}$ in front of $\sqrt{\varepsilon}$. Since feature hashing satisfies (1) whenever $\|x\|_\infty \leq \nu(m, \varepsilon, \delta)\|x\|_2$, this implies that feature hashing works with a better constant than the theory suggests.

**Proof Technique** As a fundamental step in the proof of Theorem 2 we prove tight asymptotic bounds for high-order norms of the approximation factor.[2] More formally, for every $x \in \mathbb{R}^n \setminus \{0\}$ let $X(x) = |\|Ax\|_2^2 - \|x\|_2^2|$. The technical crux of our results is tight bounds on high-order moments of $X(x)$. Note that by rescaling we may restrict our focus without loss of generality to unit vectors.

**Notation 1.** *For every $m, r, k > 0$ denote*

$$\Lambda(m, r, k) = \begin{cases} \sqrt{\frac{r}{m}}, & k \geq mr \\ \max\left\{ \sqrt{\frac{r}{m}}, \frac{r^2}{k \ln^2\left(\frac{emr}{k}\right)} \right\}, & mr > k \geq \sqrt{mr} \\ \max\left\{ \sqrt{\frac{r}{m}}, \frac{r^2}{k \ln^2\left(\frac{emr}{k}\right)}, \frac{r}{k \ln\left(\frac{emr}{k^2}\right)} \right\}, & \sqrt{mr} > k \end{cases} .$$

In these notations our main technical lemmas are the following.

**Lemma 3.** *For every even $r \leq m/4$ and every unit vector $x \in \mathbb{R}^n$, $\|X(x)\|_r = O(\Lambda(m, r, \|x\|_\infty^{-2}))$.*

**Lemma 4.** *For every $k \leq n$ and even $r \leq \min\{m/4, k\}$, $\|X(x^{(k)})\|_r = \Omega\left(\Lambda(m, r, k)\right)$, where $x^{(k)} \in \mathbb{R}^n$ is the unit vector whose first $k$ entries equal $\frac{1}{\sqrt{k}}$.*

While it might seem at a glance that bounding the high-order moments of $X(x)$ is merely a technical issue, known tools and techniques could not be used to prove Lemmas 3, 4. Particularly, earlier work by Kane and Nelson [KN14, CJN18] and Freksen and Larsen [FL17] used high-order moments bounds as a step in proving probability tail bounds of random variables. The existing techniques, however, can not be adopted to bound high-order moments of $X(x)$ (see also Section 1.2), and novel approaches were needed. Specifically, our proof incorporates a novel combinatorial scheme for counting edge-labeled Eulerian graphs.

**Previous Results.** Weinberger *et al.* [WDL$^+$09] showed that whenever $m = \Omega(\varepsilon^{-2}\lg(1/\delta))$, then $\nu(m, \varepsilon, \delta) = \Omega(\varepsilon \cdot (\lg(1/\delta)\lg(m/\delta))^{-1/2})$. Dasgupta *et al.* [DKS10] showed that under similar conditions $\nu(m, \varepsilon, \delta) = \Omega(\sqrt{\varepsilon} \cdot (\lg(1/\delta)\lg^2(m/\delta))^{-1/2})$. These bounds were recently improved by Dahlgaard *et al.* [DKT17a] who showed that $\nu(m, \varepsilon, \delta) = \Omega\left(\sqrt{\varepsilon} \cdot \sqrt{\frac{\lg(1/\varepsilon)}{\lg(1/\delta)\lg(m/\delta)}}\right)$. Conversely, Kane and Nelson [KN14] showed that for the restricted case of $m = \Theta(\varepsilon^{-2}\lg(1/\delta))$, $\nu(m, \varepsilon, \delta) = O\left(\sqrt{\varepsilon} \cdot \frac{\lg(1/\varepsilon)}{\lg(1/\delta)}\right)$, which matches the bound in Theorem 2 if, in addition, $\lg(1/\varepsilon) \leq \sqrt{\lg(1/\delta)}$.

## 1.2 Related Work

The CountSketch scheme, presented by Charikar *et al.* [CCF04], was shown to satisfy (1) by Thorup and Zhang [TZ12]. The scheme essentially samples $O(\lg(1/\delta))$ independent copies of a feature hashing matrix with $m = O(\varepsilon^{-2})$ rows, and applies them all to $x$. The estimator for $\|x\|_2^2$ is then given by computing the median norm over all projected vectors. The CountSketch scheme thus constructs a sketching matrix $A$ such that every column has $O(\lg(1/\delta))$ non-zero entries. However, this construction does not provide a norm-preserving embedding into a Euclidean space (that is, the estimator of $\|x\|_2^2$ cannot be represented as a norm of $Ax$), which is essential for some applications such as nearest-neighbor search [HIM12].

Kane and Nelson [KN14] presented a simple construction for the so-called sparse Johnson Linden-strauss transform. This is a distribution of $m \times n$ matrices, for $m = \Theta(\varepsilon^{-2}\lg(1/\delta))$, where every column has $s$ non-zero entries, randomly chosen from $\{-1, 1\}$. Note that if $s = 1$, this distribution yields the feature hashing one. Kane and Nelson showed that for $s = \Theta(\varepsilon m)$ this construction satisfies (1). Recently, Cohen *et al.* [CJN18] presented two simple proofs for this result. While their proof methods give (simple) bounds for high-order moments similar to those in Lemmas 3 and 4, they rely heavily on the fact that $s$ is relatively large. Specifically, for $s = 1$ the bounds their method or an extension thereof give are trivial.

## 2 Bounding $\nu(m, \varepsilon, \delta)$

In this section we prove the principal part of Theorem 2, assuming Lemmas 3 and 4, whose proof is deferred to the full version of the paper. Formally we prove the following.

**Theorem 5** (Main Part of Theorem 2). *There exist constants $\hat{C} > 0$ such that for every $\varepsilon, \delta \in (0, 1)$ and $m \in \mathbb{N}^+$, if $\frac{\hat{C}\lg\frac{1}{\delta}}{\varepsilon^2} \leq m < \frac{2}{\varepsilon^2\delta}$ then*

$$\nu(m, \varepsilon, \delta) = \Theta\left(\sqrt{\varepsilon} \cdot \min\left\{\frac{\lg\frac{\varepsilon m}{\lg\frac{1}{\delta}}}{\lg\frac{1}{\delta}}, \sqrt{\frac{\lg\frac{\varepsilon^2 m}{\lg\frac{1}{\delta}}}{\lg\frac{1}{\delta}}}\right\}\right) .$$

Fix $\varepsilon, \delta \in (0, 1)$ and an integer $m$. From Lemmas 3 and 4 there exist $C_1, C_2 > 0$ such that for every $r, k$, if $r \leq m/4$ then for every unit vector $x$, $\|X(x)\|_r \leq 2^{C_2}\Lambda(m, r, k)$. Moreover, if $r \leq k$ then

$$2^{-C_1}\Lambda(m, r, k) \leq \|X(x^{(k)})\|_r \leq 2^{C_2}\Lambda(m, r, k) .$$

Note that in addition $\Lambda(m, 2r, k) \leq 4\Lambda(m, r, k)$. Denote $\hat{C} = 2^{C_2+2}$, and $C = 2C_1 + 2C_2 + 5$.

For the rest of the proof we assume that $\frac{\hat{C}\log\frac{1}{\delta}}{\varepsilon^2} \leq m < \frac{2}{\varepsilon^2\delta}$, and we start by proving a lower bound on $\nu$.

**Lemma 6.** $\nu(m, \varepsilon, \delta) = \Omega\left(\min\left\{\frac{\sqrt{\varepsilon}}{\lg\frac{1}{\delta}}\lg\frac{\varepsilon m}{\lg\frac{1}{\delta}}, \sqrt{\frac{\varepsilon\lg\frac{\varepsilon^2 m}{\lg\frac{1}{\delta}}}{\lg\frac{1}{\delta}}}\right\}\right)$.

*Proof.* Let $r = \lg\frac{1}{\delta}$, let $x \in \mathbb{R}^n$ be a unit vector such that $\|x\|_\infty \leq \min\left\{\frac{\sqrt{\varepsilon}\ln\frac{e\varepsilon m}{r}}{\sqrt{2^{C_2}er}}, \sqrt{\frac{\varepsilon\lg\frac{e\varepsilon^2 m}{r}}{2^{C_2}er}}\right\}$, and let $k := \frac{1}{\|x\|_\infty^2} \geq \max\left\{\frac{2^{C_2}er^2}{\varepsilon\ln^2\frac{e\varepsilon m}{r}}, \frac{2^{C_2}er}{\varepsilon\lg\frac{e\varepsilon^2 m}{r}}\right\}$. If $k \leq mr$, then since $\frac{r^2}{k\ln^2\frac{emr}{k}}$ is convex as a

function of $k \in \left[\frac{2^{C_2}er^2}{\varepsilon \ln^2 \frac{e\varepsilon m}{r}}, mr\right]$ then

$$2^{C_2}\frac{r^2}{k \ln^2 \frac{emr}{k}} \le \max\left\{\frac{r}{m}, \left(\frac{\varepsilon \ln^2 \frac{e\varepsilon m}{r}}{e \ln^2 \frac{e\varepsilon m \ln^2 \frac{e\varepsilon m}{r}}{r}}\right)\right\} < \varepsilon/2 \ .$$

Moreover, if $k \le \sqrt{mr}$ then since $\frac{r}{k \ln \frac{emr}{k^2}}$ is convex as a function of $k \in \left[\frac{2^{C_2}er}{\varepsilon \ln \frac{e\varepsilon^2 m}{r}}, \sqrt{mr}\right]$, then

$$2^{C_2}\frac{r}{k \ln \frac{emr}{k^2}} \le \max\left\{\sqrt{\frac{r}{m}}, \left(\frac{\varepsilon \ln \frac{e\varepsilon^2 m}{r}}{e \ln \frac{\varepsilon^2 m \ln^2 \frac{e\varepsilon^2 m}{r}}{r}}\right)\right\} \le \varepsilon/2 \ .$$

Since clearly, $\sqrt{\frac{2^{2C_2}r}{m}} \le \varepsilon/2$, then by Lemma 3 we have $\|X(x)\|_r^r \le (\varepsilon/2)^r$, and thus

$$\Pr\left[\ \left|\|Ax\|_2^2 - 1\right| > \varepsilon\ \right] = \Pr\left[\ (X(x))^r > \varepsilon^r\ \right] \le 2^{-r} = \delta \ . \tag{2}$$

Hence $\nu(m,\varepsilon,\delta) \ge \min\left\{\frac{\sqrt{\varepsilon}\ln \frac{e\varepsilon m}{r}}{\sqrt{2^{C_2}er}}, \sqrt{\frac{\varepsilon \lg \frac{e\varepsilon^2 m}{r}}{2^{C_2}er}}\right\} = \Omega\left(\min\left\{\frac{\sqrt{\varepsilon}}{\lg\frac{1}{\delta}}\lg\frac{\varepsilon m}{\lg\frac{1}{\delta}}, \sqrt{\frac{\varepsilon \lg \frac{\varepsilon^2 m}{\lg\frac{1}{\delta}}}{\lg\frac{1}{\delta}}}\right\}\right).$ $\quad\square$

**Lemma 7.** $\nu(m,\varepsilon,\delta) = O\left(\min\left\{\frac{\sqrt{\varepsilon}}{\lg\frac{1}{\delta}}\lg\frac{\varepsilon m}{\lg\frac{1}{\delta}}, \sqrt{\frac{\varepsilon \lg \frac{\varepsilon^2 m}{\lg\frac{1}{\delta}}}{\lg\frac{1}{\delta}}}\right\}\right).$

To this end, let $r = \frac{1}{C}\lg\frac{1}{\delta}$, and denote

$$t = \min\left\{\frac{\sqrt{e\varepsilon}}{r}\ln\frac{e\varepsilon m}{r}, \sqrt{\frac{e\varepsilon \ln\frac{e\varepsilon^2 m}{r}}{r}}\right\} = O\left(\min\left\{\frac{\sqrt{\varepsilon}}{\lg\frac{1}{\delta}}\lg\frac{\varepsilon m}{\lg\frac{1}{\delta}}, \sqrt{\frac{\varepsilon \lg \frac{\varepsilon^2 m}{\lg\frac{1}{\delta}}}{\lg\frac{1}{\delta}}}\right\}\right) \ .$$

Assume first that $t \le \frac{1}{\sqrt{r}}$, and let $k = \frac{1}{t^2}$. We will show that $\mathbb{E}\left[\left(X(x^{(k)})\right)^r\right] \ge 2\varepsilon^r$. Since $t \le \frac{1}{\sqrt{r}}$, then $k \ge r$. If $\frac{\sqrt{e\varepsilon}}{r}\ln\frac{e\varepsilon m}{r} \le \sqrt{\frac{e\varepsilon \ln\frac{e\varepsilon^2 m}{r}}{r}}$, then $k = \frac{r^2}{e\ln^2\frac{e\varepsilon m}{r}}$. Since $\frac{e\varepsilon m}{r} > e$, then $k \le mr$. Therefore

$$\mathbb{E}\left[\left(X(x^{(k)})\right)^r\right] = \|X(x^{(k)})\|_r^r \ge \left(\frac{r^2}{k \ln^2 \frac{emr}{k}}\right)^r = \left(\frac{e\varepsilon \ln^2 \frac{e\varepsilon m}{r}}{\ln^2 \frac{e^2\varepsilon m \ln^2 \frac{e\varepsilon m}{r}}{r}}\right)^r \ge 2\varepsilon^r \ .$$

Otherwise, $k = \frac{r}{e\varepsilon \ln\frac{e\varepsilon^2 m}{r}}$. Moreover, since $\frac{\varepsilon^2 m}{r} > 1$, then $k \le r/\varepsilon \le \sqrt{mr}$. Therefore

$$\mathbb{E}\left[\left(X(x^{(k)})\right)^r\right] = \|X(x^{(k)})\|_r^r \ge \left(\frac{r}{k \ln \frac{emr}{k^2}}\right)^r = \left(\frac{e\varepsilon \ln \frac{e\varepsilon^2 m}{r}}{\ln \frac{e^3\varepsilon^2 m \ln^2 \frac{e\varepsilon^2 m}{r}}{r}}\right)^r \ge 2\varepsilon^r \ .$$

Applying the Paley-Zygmund inequality we get that

$$\Pr\left[\ \left|\|Ax^{(k)}\|_2^2 - 1\right| > \varepsilon\ \right] \ge \Pr\left[\left(X(x^{(k)})\right)^r > \frac{1}{2}\mathbb{E}[X(x^{(k)})^r]\right] \ge \frac{1}{4}\left(\frac{2^{-C_1}\Lambda(m,r,k)}{2^{C_2}\Lambda(m,2r,k)}\right)^{2r} \ge \delta \ .$$

Therefore $\nu(m,\varepsilon,\delta) \le \|x^{(k)}\|_\infty = t$.

Assume next that $\frac{1}{\sqrt{r}} < t < \sqrt{\frac{\varepsilon}{4}}$, and note that since $\frac{\sqrt{e\varepsilon}}{r}\ln\frac{e\varepsilon m}{r} \ge \frac{1}{\sqrt{r}}$, then $m > e^{\sqrt{r/(e\varepsilon)}}$, and since $\sqrt{\frac{e\varepsilon \ln\frac{e\varepsilon^2 m}{r}}{r}} \ge \frac{1}{\sqrt{r}}$ then $m > e^{1/(e\varepsilon)}$. Let $k = \frac{1}{t^2}$, and consider independent $h \in_R [n] \to [m]$, and $\sigma = (\sigma_1,\ldots,\sigma_m) \in_R \{-1,1\}^m$. Let $y \in \mathbb{R}^n$ be defined as follows. For every $j \in [n]$, $y_j = x_j^{(k)}$ if and only if $h(j) = 1$, and $y_j = 0$ otherwise. Denote $z = x^{(k)} - y$. Then $\|x^{(k)}\|_2^2 = \|y\|_2^2 + \|z\|_2^2$,

and moreover, $\|Ax^{(k)}\|_2^2 = \|Ay\|_2^2 + \|Az\|_2^2$, where $A = A(h, \sigma)$. Let $\mathcal{E}_{first}$ denote the event that $|h^{-1}(\{1\})| = 2\sqrt{\varepsilon k}$, and that for all $j \in [n]$, if $h(j) = 1$ then $\sigma_j = 1$, and let $\mathcal{E}_{rest}$ denote the event that $\left|\|Az\|_2^2 - \|z\|_2^2\right| < \varepsilon\|z\|_2^2$. By Chebyshev's inequality, $\Pr[\mathcal{E}_{rest} \mid \mathcal{E}_{first}] = \Omega(1)$. Note that if $k = \frac{r^2}{e\varepsilon \ln^2 \frac{e\varepsilon m}{r}}$, then since $m > \max\{e^{1/e\varepsilon}, e^{\sqrt{r}}\}$ we get

$$2\sqrt{\varepsilon k} \le \frac{\lg\frac{1}{\delta}}{C\sqrt{e}\ln\frac{e\varepsilon m}{r}} \le \frac{\lg\frac{1}{\delta}}{C\sqrt{e}(\ln em - \ln\frac{1}{\varepsilon} - \ln r)} \le \frac{\lg\frac{1}{\delta}}{C\sqrt{e}(\ln m - 3\ln\ln m)} \le \frac{\lg\frac{1}{\delta}}{2\ln m} ,$$

and otherwise, $k = \frac{r}{e\varepsilon \ln \frac{e\varepsilon^2 m}{r}}$, and

$$2\sqrt{\varepsilon k} \le \varepsilon k = \frac{\lg\frac{1}{\delta}}{eC\ln\frac{e\varepsilon^2 m}{r}} = \frac{\lg\frac{1}{\delta}}{eC(\ln em - 2\ln\frac{1}{\varepsilon} - \ln r)} \le \frac{\lg\frac{1}{\delta}}{eC(\ln em - 4\ln\ln m)} \le \frac{\lg\frac{1}{\delta}}{2\ln m} .$$

Therefore for small enough $\varepsilon$,

$$\Pr[\mathcal{E}_{first}] = \binom{k}{2\sqrt{\varepsilon k}} \cdot \left(\frac{1}{m}\right)^{2\sqrt{\varepsilon k}} \cdot \left(1 - \frac{1}{m}\right)^{k - 2\sqrt{\varepsilon k}} \cdot 2^{-2\sqrt{\varepsilon k}}$$

$$\ge \left(\frac{1}{m}\right)^{2\sqrt{\varepsilon k}} \cdot \left(1 - \frac{1}{m}\right)^r \cdot 2^{-r} \ge \left(\frac{1}{m}\right)^{2\sqrt{\varepsilon k}} \cdot 2^{-\frac{2}{C}\lg\frac{1}{\delta}} \ge \delta^{3/4} .$$

Thus for small enough $\delta$, $\Pr[\mathcal{E}_{first} \wedge \mathcal{E}_{rest}] \ge \delta$. Conditioned on $\mathcal{E}_{first} \wedge \mathcal{E}_{rest}$ we get that

$$\|Ax^{(k)}\|_2^2 = \|Ay\|_2^2 + \|Az\|_2 \ge \frac{4\varepsilon k}{k} + (1-\varepsilon)\|z\|_2^2 = 4\varepsilon + (1-\varepsilon) \cdot \frac{k - 2\sqrt{\varepsilon k}}{k} \ge 4\varepsilon + (1-\varepsilon)^2 > 1 + \varepsilon ,$$

where the inequality before last is due to the fact that $k \ge \frac{4}{\varepsilon}$. Therefore $\nu(m, \varepsilon, \delta) \le \|x^{(k)}\|_\infty = t$.

Finally, assume $t > \sqrt{\frac{\varepsilon}{4}}$. Since $\sqrt{\frac{e\varepsilon \ln \frac{e\varepsilon^2 m}{r}}{r}} \ge t > \sqrt{\frac{\varepsilon}{4}}$, we get that $m \ge \frac{r}{e\varepsilon^2} e^{r/(4e)} \ge \frac{r}{e\varepsilon^2 \delta^{1/(4eC)}}$. Let $k = \frac{2}{\varepsilon}$. Consider independent $h \in_R [n] \to [m]$, and $\sigma = (\sigma_1, \ldots, \sigma_m) \in_R \{-1, 1\}^m$, and let $A = A(h, \sigma)$. Let $\mathcal{E}_{col}$ denote the event that there are $j \ne \ell \in [k]$ such that for every $p \ne q \in [k]$, $h(p) = h(q)$ if and only if $\{p, q\} = \{j, \ell\}$. Then for small enough $\varepsilon, \delta$,

$$\Pr[\mathcal{E}_{col}] = \binom{k}{2} \cdot \frac{1}{m} \cdot \prod_{j \in [k-1]} \left(1 - \frac{j}{m}\right) \ge \frac{k^2}{2m} \cdot (1 - \varepsilon/2) \cdot \left(1 - \frac{k}{m}\right)^k$$

$$\ge \frac{k^2}{2m} \cdot (1 - \varepsilon/2) \cdot \left(1 - \frac{k^2}{m}\right) \ge 2\delta \cdot (1 - \varepsilon/2) \cdot \left(1 - 4e\delta^{1/(4Ce)}\right) \ge \delta .$$

Conditioned on $\mathcal{E}_{col}$ we get that $\left|\|Ax^{(k)}\|_2^2 - 1\right| = \frac{2}{k} = \varepsilon$. Therefore $\nu(m, \varepsilon, \delta) \le \sqrt{\frac{\varepsilon}{2}} \le O(t)$. This completes the proof of Lemma 7, and thus of Theorem 5.

## 3 Empirical Analysis

We complement our theoretical bounds with experimental results on both real and synthetic data. The goal of the experiments is to give bounds on some of the constants hidden in the main theorem. Our synthetic-data experiments show that for $\frac{4\lg\frac{1}{\delta}}{\varepsilon^2} \le m < \frac{2}{\varepsilon^2\delta}$ the constant inside the $\Theta$-notation in Theorem 2 is at least $0.725$ except for very sparse vectors ($\|x\|_0 \le 7$), where the constant is at least $0.6$. Furthermore, we confirm that $\nu(m, \varepsilon, \delta) = 1$ when $m \ge \frac{2}{\varepsilon^2\delta}$ and that there exists data points where $\nu(m, \varepsilon, \delta) < 1$ while $m = \frac{2-\gamma}{\varepsilon^2\delta}$, for some small $\gamma$. In addition, for the real-world data we tested feature hashing on, the constant is at least $1.1$ or $0.8$, based on the data set.

### 3.1 Experiment Setup and Analysis

To arrive at the results, we ran experiments and analysed the data in several phases. In the first phase we varied the target dimension $m$ over exponentially spaced values in the range $[2^4, 2^{14}]$, and a parameter $k$ which controls the ratio between the $\ell_\infty$ and the $\ell_2$ norm. The values of $k$ varied over

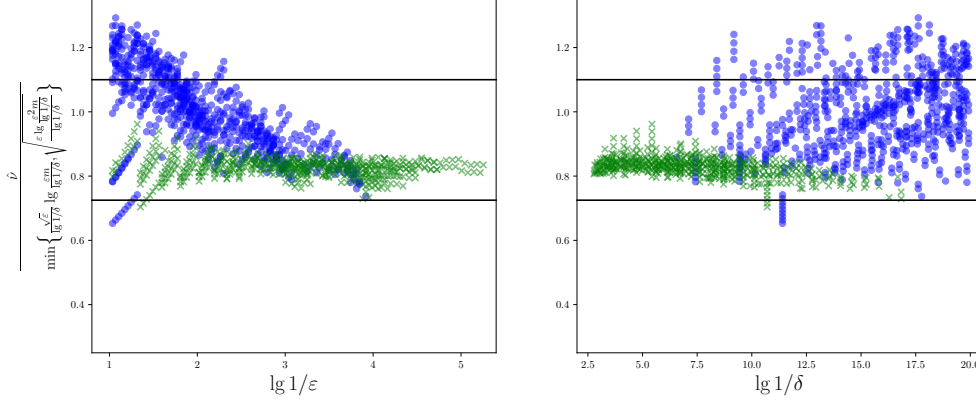

Figure 1: The plot shows the ratio between $\hat{\nu}$ values and the theoretical bound, abbreviated here as $\min\{\textsf{left}, \textsf{right}\}$. This ratio corresponds to the constant in the $\Theta$-notation in Theorem 2. The points are marked with blue circles if $\textsf{left} < \textsf{right}$, and with green $\times$'s otherwise. The horizontal line at 0.725 is there to ease comparisons with Figure 2, while the line at 1.1 helps in comparing against real world data (Figure 4 and Figure 5).

exponentially spaced values in the range $[2^1, 2^{13}]$. Then for all $m$ and $k$, we generated $2^{24}$ vectors $x$ with entries in $\{0, 1\}$ such that $\|x\|_2 = \sqrt{k}\|x\|_\infty$, and for any given $m$ and $k$ the supports of the vectors were pairwise disjoint. We then hashed the generated vectors using feature hashing, and recorded the $\ell_2$ norm of the embedded vectors.

The second phase then calculated the distortion between the original and the embedded vectors, and computed the error probability $\hat{\delta}$. Loosely speaking, $\hat{\delta}(m, k, \varepsilon)$ is the ratio of the $2^{24}$ vectors for a given $m$ and $k$ that have distortion greater than $\varepsilon$. Formally, $\hat{\delta}$ is calculated using the following formula

$$\hat{\delta}(m, k, \varepsilon) = \frac{\left|\left\{x : \|x\|_2 = \sqrt{k}\|x\|_\infty, \left|\|A_m x\|_2^2 - \|x\|_2^2\right| \geq \varepsilon\|x\|_2^2\right\}\right|}{\left|\left\{x : \|x\|_2 = \sqrt{k}\|x\|_\infty\right\}\right|},$$

where $\varepsilon$ was varied over exponentially spaced values in the range $[2^{-10}, 2^{-1}]$. Note that $\hat{\delta}$ tends to the true error probability as the number of vectors tends to infinity. Computing $\hat{\delta}$ yielded a series of 4-tuples $(m, k, \varepsilon, \hat{\delta})$ which can be interpreted as given target dimension $m$, $\ell_\infty/\ell_2$ ratio $1/\sqrt{k}$, distortion $\varepsilon$, we have measured that the failure probability is at most $\hat{\delta}$.

In the third phase, we varied $\delta$ over exponentially spaced values in the range $[2^{-20}, 2^0]$, and calculated a value $\hat{\nu}$. Intuitively, $\hat{\nu}(m, \varepsilon, \delta)$ is the largest $\ell_\infty/\ell_2$ ratio such that for all vectors having at most this $\ell_\infty/\ell_2$ ratio the measured error probability $\hat{\delta}$ is at most $\delta$. Formally,

$$\hat{\nu}(m, \varepsilon, \delta) = \max\left\{\frac{1}{\sqrt{k}} : \forall k' \geq k, \hat{\delta}(m, k', \varepsilon) \leq \delta\right\}.$$

Note once more that $\hat{\nu}$ tends to the true $\nu$ value as the number of vectors tends to infinity. To find a bound on the constant of the $\Theta$-notation in Theorem 2, we truncated data points that did not satisfy $\frac{4 \lg \frac{1}{\delta}}{\varepsilon^2} \leq m < \frac{2}{\varepsilon^2 \delta}$, and for the remaining points we plotted $\hat{\nu}$ over the theoretical bound in Figure 1. From this plot we conclude that the constant is at least 0.6 on the large range on parameters we tested. However, the smallest values seem to be outliers and come from a combination of very sparse vectors ($k = 7$) and high target dimension ($m = 2^{14}$). For the rest of the data points the constant is at least 0.725. While there are data points where the constant is larger (i.e. feature hashing performs better), there are data points close to 0.725 over the entire range of $\varepsilon$ and $\delta$.

In Figure 2 we show that we indeed need both terms in the minimum in Theorem 2, by plotting the measured $\hat{\nu}$ values over both terms in the minimum in the theoretical bound separately. For both terms there are points whose value is significantly below 0.725.

To find a bound on $m$ where $\hat{\nu}(m, \varepsilon, \delta) = 1$ we took the untruncated data and recorded the maximal $\hat{\delta}$ for each $m$ and $\varepsilon$. We then plotted $m\varepsilon^2\hat{\delta}$ in Figure 3. From Figure 3 it is clear that $\hat{\nu}(m, \varepsilon, \delta) = 1$

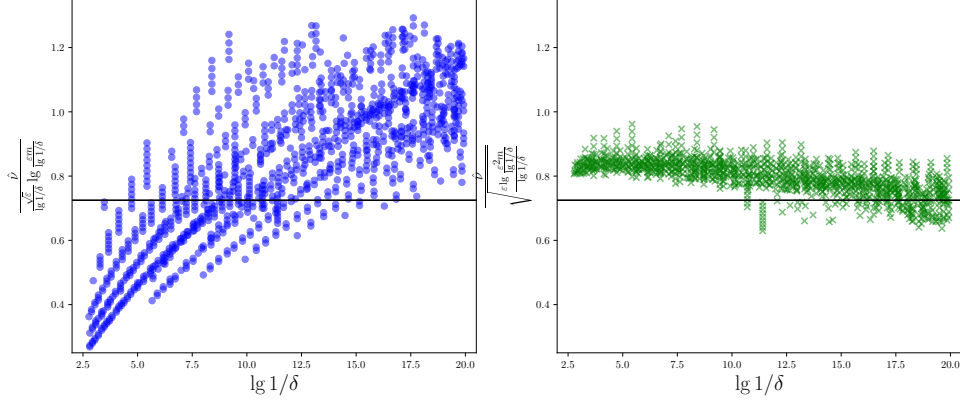

Figure 2: This plot shows the measured $\hat{\nu}$ values over each of the two terms in the minimum in the theoretical bound (abbreviated here): $\min\{\text{left}, \text{right}\}$. In the left subfigure the $y$-axis of the blue circles is $\frac{\hat{\nu}}{\text{left}}$, while the $y$-axis of the green $\times$'s in the right subfigure is $\frac{\hat{\nu}}{\text{right}}$. Note that the $x$-axis (values of $\lg(1/\delta)$) is the same in both subfigures, and the same as in the right subfigure of Figure 1. As in Figure 1, the horizontal line at $0.725$ is there to ease comparison between the figures.

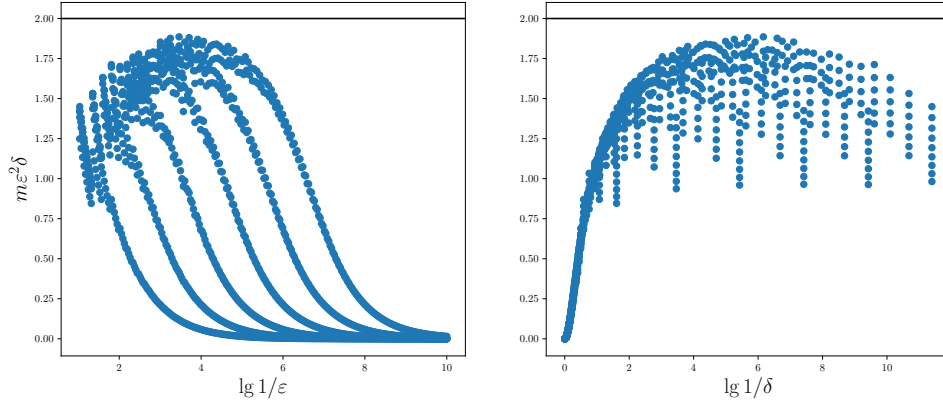

Figure 3: This plot shows the constant where $\hat{\nu}(m, \varepsilon, \delta)$ becomes 1. The theory states that if $2 \leq m\varepsilon^2\delta$ then $\hat{\nu}(m, \varepsilon, \delta) = 1$. The distinct curves in the left plot correspond to distinct values of $m$.

when $m \geq \frac{2}{\varepsilon^2\delta}$. Furthermore, the figure also shows that there are data points where $\hat{\nu}(m, \varepsilon, \delta) < 1$ while $m = \frac{2-\gamma}{\varepsilon^2\delta}$, for some small $\gamma$. Therefore we conclude the bound $m \geq \frac{2}{\varepsilon^2\delta}$ is tight.

### 3.2 Real-World Data

We also ran experiments on real-world data, namely bag-of-words representations of 1500 NIPS papers with stopwords removed [DKT17b, New08]. We ran experiments on this data set both with and without preprocessing with the common logarithmic term frequency - inverse document frequency (tf-idf). These experiments were executed and analysed similarly to the synthetic experiments described above, except for a few changes. First, in order to explore any meaningful $\delta$ values we hashed each vector $2^{20}$ times. In this way, iterating over the original vectors in the real world experiments plays a similar role to iterating over the $k$ values in the synthetic experiments. Secondly, in these experiments $m$ ranged over values in $[2^4, 2^{12}]$.

The results of these experiments can be seen in Figure 4 and Figure 5, from which we conclude that feature hashing performs even better on the real-world data we tested compared to the synthetic data, as the Theorem 2 constant is always above $1.1$ and $0.8$ with and without tf-idf, respectively. Furthermore, for the vast majority of data points have a constant around or above $1.2$.

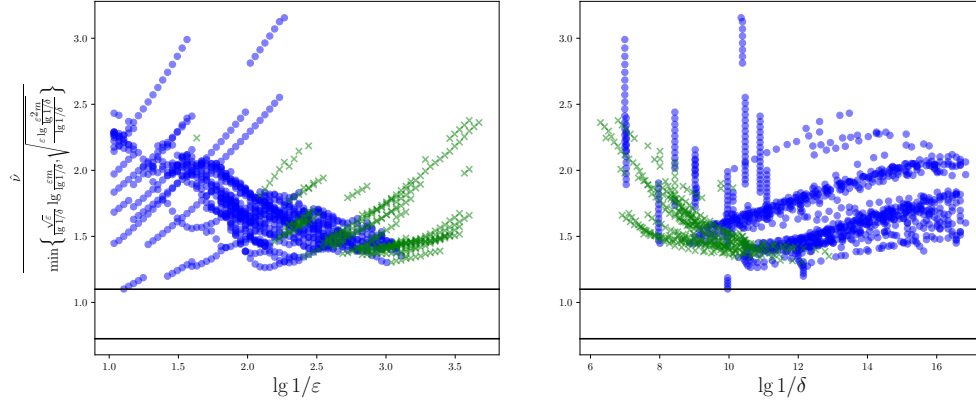

Figure 4: This plot has a similar structure to Figure 1, but is based on the NIPS dataset preprocessed with tf-idf. This plot shows the measured $\hat{\nu}$ values over the theoretical bound (abbreviated here): $\min\{\mathsf{left},\mathsf{right}\}$. This ratio corresponds to the constant in the $\Theta$-notation in Theorem 2. The points are marked with blue circles if $\mathsf{left} < \mathsf{right}$, otherwise they are marked with green $\times$'s. The horizontal lines at 0.725 and 1.1 are there to ease comparisons with Figure 1.

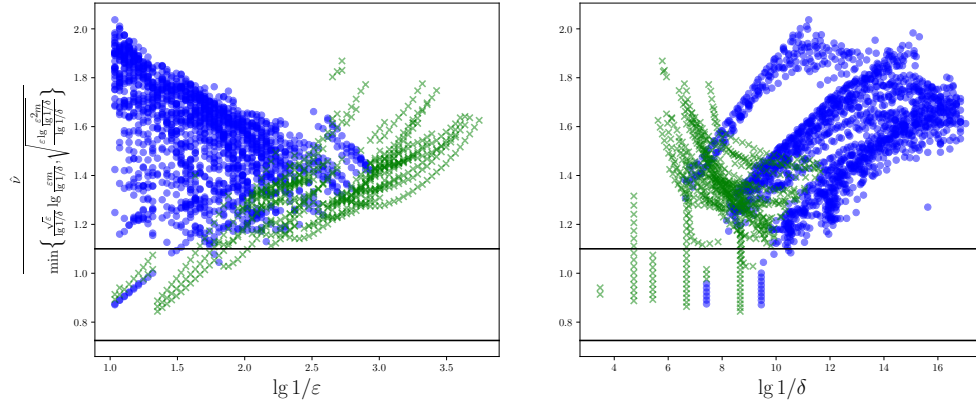

Figure 5: This plot has a similar structure to Figure 1, but is based on the NIPS dataset without tf-idf preprocessing. This plot shows the measured $\hat{\nu}$ values over the theoretical bound (abbreviated here): $\min\{\mathsf{left},\mathsf{right}\}$. This ratio corresponds to the constant in the $\Theta$-notation in Theorem 2. The points are marked with blue circles if $\mathsf{left} < \mathsf{right}$, otherwise they are marked with green $\times$'s. The horizontal lines at 0.725 and 1.1 are there to ease comparisons with Figure 1.

### 3.3 Implementation Details

As random number generators, we used degree 20 polynomials modulo the Mersenne prime $2^{61} - 1$, where the coefficients were random data from random.org. The random data was independent between experiments with diffent values of $m$, between synthetic and real world experiments, and between the random number generator used for vector generation and hashing.

Feature hashing was done using double tabulation hashing [Tho14] on 64 bit numbers. The tables in our implementation of double tabulation hashing were filled with numbers from the aforementioned random number generator. Double tabulation hashing has been proven to behave fully randomly with high probability [DKRT15].

In order to efficiently do the $2^{20}$ hashings per vector for the real world data, we utilised the high independence of double tabulation hashing. Let $d$ be the original dimension of the vectors. We blew up the source dimension to $2^{20}d$, and at the $i$th rehash we shifted the coordinates of the original vector $i \cdot d$ places to the right.

**Acknowledgments**

This work was supported by a Villum Young Investigator Grant.

## Footnotes

* All authors contributed equally, and are presented in alphabetical order.

[2] Given a random variable $X$ and $r > 0$, the $r$th norm of $X$ (if exists) is defined as $\|X\|_r := \sqrt[r]{\mathbb{E}(|X|^r)}$.

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
