[Reviews · NeurIPS 2018]

Reviewer 1



The authors carry out a fine-grained analysis of locality sensitive hashing, and prove data-dependent bounds in order to provide a better understanding of how much random compression can be applied to data in realistic settings, while still preserving Johnson-Lindenstrauss guarantees on the compressed data. This is useful to know, since existing lower bounds on the projected dimension generally proceed by finding the minimal projection dimension required to compress the worst case (e.g. data lying at the vertices of the \ell_1 ball or some similarly structured scenario) but in practice observed data frequently have much more forgiving structure. The paper is interesting and, as far as I can see, correct although I did not check *all* the details thoroughly. Although there is nothing really wrong with this paper - and I commend the authors for their work - I do however have two reasons why I think it may not be suitable for NIPS: The first is that I think this work is likely of more interest to a databases, KDD, or theoretical CS audience and therefore NIPS is perhaps not the best venue to gain this work the widest audience of interested readers. The authors clearly feel otherwise, but this is my well-intended advice. The second, more serious, concern I have is that - as a purely theoretical paper - I think that the main contributions here are really the proof techniques in the supplementary material, and I think there is not enough room to include even a reasonable sketch of these within the NIPS page limit. That is, I don't think the contributions in the manuscript are enough for NIPS but, notwithstanding my comment above, I do think that the longer paper in the supplementary paper could have been. Since NIPS papers should be self-contained I think that is a problem. FWIW I think the supplementary version of this paper is clearly of a publishable standard at a very good venue, and believe the authors would be helping themselves by either disseminating their work as a journal paper or else submitting to a conference with a more generous page limit (e.g. ECML-PKDD 16 pages in LNCS style) or a tighter style (e.g. KDD 9 pages inc. refs. in ACM style). Please note also that I do not object to this paper because it is theoretical: I think that theory papers are important contributions to all of the serious conferences, and personally I would be happy to see more of them. However in my own experience it is not always easy to balance providing enough detail about how the results were obtained with substantial outcomes in a short conference paper, and here I don't think the authors have found that balance. Addendum: The authors did not address my comments in rebuttal, but checking post-review I see that the supp mat version is available on the ArXiv. I prefer that version to the NIPS version a lot, and encourage the authors (if they need any encouragement) to link to it from their NIPS submission should it be accepted. Since a guarantee of the details absent in the NIPS version in some archival form was my main concern here, I have accordingly raised my rating to "accept".

Reviewer 2



The paper provides tight bounds on the performance of dimension reduction using (very) sparse matrices on non-worst case input. In general, the performance of dimension reduction using linear maps has been analyzed but the worst case instances are vectors that are extremely sparse. In previous work by Weinberger et al., they focused on the setting where the maximum coordinate value is much smaller than the vector length i.e. the vector is not too sparse. This is motivated by the fact that input vectors e.g. document-word incident matrices are not pathological and they are fairly spread-out. The speed of dimension reduction is proportional to the sparsity of the embedding matrix. In the worst case, the embedding matrix has to be reasonably dense. However for the setting of Weinberger et al., one can use even matrices with 1 non-zero per column, which is the minimum possible sparsity. This paper gives tight bounds for the relation between how well-distributed the input vector x is, the target dimensions, the distortion of the length, and the failure probability. As this tradeoff involves many parameters, the precise relation is quite complicated. The result follows from a careful analysis of the high moment of the error |||Ax||-||x|||. The basic approach is classical: one expands the expression for the high moment and count the number of terms. The terms can be associated with graphs and the problem becomes counting graphs (with the indices of the coordinates as vertices). Getting accurate bounds on the number of graphs is complicated and I have not checked the details here. Overall, this paper establishes tight bounds for a complicated tradeoff for an interesting problem. A downside is that the experiments are done only with synthetic vectors. It would be interesting to see if the theory here is predictive of the performance on real datasets such as the settings considered by Weinberger et al.

Reviewer 3



I really like the paper. The authors provide the detailed analysis of the hashing trick (a.k.a. sparse random projection, a.k.a. count-sketch). Namely, how does the number of rows depends on the "spread" of the mass of the vector, desired distortion and the probability of success? The paper figures out the answer that is sharp up to a constant for all of the regimes! The proof follows by now standard technique of bounding higher moments, but to accomplish this, one needs hairy combinatorial considerations. Overall, I think the result is more interesting than the proof, but the result is great! Two minor comments: * Hashing trick was used for the nearest neighbor search (see https://arxiv.org/abs/1509.02897). This should be cited along with other applications. * The title of Section 3 has a typo.